# Angiogenesis of Avascular Necrosis of the Femoral Head: A Classic Treatment Strategy

**DOI:** 10.3390/biomedicines12112577

**Published:** 2024-11-11

**Authors:** Ping Wang, Wenkai Shao, Yuxi Wang, Bo Wang, Xiao Lv, Yong Feng

**Affiliations:** 1Department of Orthopedics, Union Hospital, Tongji Medical College, Huazhong University of Science and Technology, Wuhan 430022, China; m202276006@hust.edu.cn (P.W.); wenkaishao@163.com (W.S.); husterwyx29@163.com (Y.W.); xlyu@hust.edu.cn (X.L.); 2Department of Rehabilitation, Wuhan No. 1 Hospital, Tongji Medical College, Huazhong University of Science and Technology, Wuhan 430030, China; wangbob2019@126.com

**Keywords:** avascular necrosis of the femoral head, angiogenesis, endothelial cell metabolism

## Abstract

Avascular necrosis of the femoral head (ANFH) is a type of osteonecrosis due to the cessation of blood supply, characterized by persistent local pain and collapse of the joint. The etiology of ANFH is multifaceted, and while its precise pathogenesis remains elusive, it is currently widely believed that the femoral head is highly dependent on the vascular system. A large number of studies have shown that vascular injury is the initial factor in the onset of ANFH. In this review, we briefly introduced the process of angiogenesis and the blood supply to the femoral head, with a focus on summarizing the existing research on promoting angiogenesis for the treatment of ANFH. We conclude that providing alternative pathways through angiogenesis to resolve the problem of the obstructed free flow of the blood is an important means of treating ANFH. Moreover, we also looked forward to the mechanism of endothelial metabolism, which has not yet been studied in femoral head necrosis models, providing potential strategies for more effective use of angiogenesis for the treatment of femoral head necrosis.

## 1. Introduction

Avascular necrosis of the femoral head (ANFH) is a debilitating bone disorder characterized by persistent pain, which poses a significant challenge to treat [1]. In the United States, between 20,000 and 30,000 new cases are reported annually [2]. The etiology of ANFH is multifaceted and still unknown; over 80% of cases are linked to the use of corticosteroids and excessive alcohol use, which has attracted significant study interest [3,4]. In addition, disease factors such as systemic lupus erythematosus, fracture, dislocation, and sickle cell disease, as well as treatments such as chemotherapy, radiation, and hip surgery, have been identified as possible risk factors associated with the progression of ANFH [5,6,7,8,9]. Although its exact pathophysiological mechanism remains unclear, it is currently widely believed that the femoral head is highly dependent on the vascular system. Ischemia resulting from various conditions is considered the primary pathogenesis. The destruction of subchondral bone structure, femoral head necrosis and collapse, an inflammatory cascade, and associated symptoms might result from an insufficient local blood supply [10]. Three possible mechanisms have been put forth thus far: vascular interruption, vascular occlusion, and extravascular intraosseous compression [11]. Angiogenesis usually refers to the complex process of blood vessels sprouting from pre-existing ones and generating new ones, which is strictly regulated by various angiogenic and inhibitory factors in the microenvironment [12]. Numerous studies have demonstrated the tight relationship between angiogenesis and the occurrence and progression of many diseases, as well as the fact that abnormal and impaired angiogenesis are signs of the occurrence and development of many diseases [13,14]. A large number of studies have shown that vascular damage is the initial factor in the onset of ANFH [15,16], and impaired angiogenesis is a key mechanism in the occurrence of ANFH [17].

Researchers have long acknowledged the significance of angiogenesis, and more recently, they have delved into the molecular mechanisms of angiogenesis. Based on this, they have explored pro- and anti-angiogenic drugs as possible therapeutic avenues, consistently providing new viewpoints and means for the treatment of related diseases. In ANFH, similar studies have shown that promoting angiogenesis through various pathways can be used to treat this disease [18,19].

In this review, we describe the basic process of angiogenesis and the pathological events of angiogenesis damage. By exploring the complex molecular mechanisms and signaling pathways of this process, we present some possible treatment measures for intervening in angiogenesis therapy or reducing the risk of ANFH. This offers a solid foundation for future researchers to understand, research, develop, and treat ANFH from an angiogenesis standpoint.

## 2. Angiogenesis

The vascular system, which is formed by arteries, veins, and capillaries, is a closed tree-shaped structure network composed mainly of endothelial cells (ECs) and includes smooth muscle cells, fibroblasts, pericytes, and mesenchymal stromal cells close to them. The endothelial cells that make up the blood vessels are in a state of protracted quiescence and rarely proliferate. However, these endothelial cells retain the ability to form new blood vessels and can rapidly generate new vessels under specific conditions (such as in ischemic and hypoxic environments where increased blood supply is needed) to meet the body’s demands [20]. Angiogenesis refers to the process of forming new blood vessels based on pre-existing ones, that is, generating new vessels from existing capillaries or venules through the proliferation, differentiation, and migration of endothelial cells, either by sprouting or non-sprouting mechanisms. This is an “additive” growth process that goes from fewer to more. It plays a significant role in various physiological and pathological processes and is closely related to the development of various organs [21].

Angiogenesis can be subdivided into two main types: sprouting angiogenesis and intussusceptive angiogenesis (IA, aka vessel splitting or non-SA) (Figure 1) [20]. Sprouting angiogenesis refers to the fact that endothelial cells (ECs) in blood vessels are activated by pro-angiogenesis factors (such as VEGF) and differentiate into proliferating stalk cells and migrating tip cells, respectively, among which, tip cells, which are highly polarized and have low proliferation capacity, can guide the growth of nascent sprout towards areas of hypoxia and nutrient deficiency, while stalk cells proliferate to ensure sprout elongation and mediate lumen formation [22,23]; proliferating stalk cells and migrating tip cells work together to mediate the formation of new blood vessels. The characteristic of intussusceptive angiogenesis is the formation of hollow intraluminal tissue columns directly in the blood vessels, which grow and fuse along the circumference, and eventually split into two parallel blood vessels, which is also known as splitting angiogenesis [24,25,26]. Besides these two, there are some special ways that angiogenesis appears in tumors, such as vessel co-option (VC) [27], vascular mimicry (VM) [28], or cancer stem cell transdifferentiation into ECs [29], but since these are not relevant to our topic, we will not discuss them further here.

The influence of angiogenesis on the body has both good and bad sides, which has made the drug targets for pro-angiogenesis and anti-angiogenesis a hot spot that receives extensive attention from researchers, with hopes to cure some diseases by regulating angiogenesis. Next, we will focus on describing angiogenesis in ANFH.

## 3. Vascular Distribution of Femoral Head

As the longest bone in the human body, the vascularization of the femoral head is extremely high, and its mechanical strength, performance, repair, regeneration, and remodeling ability all depend on the health of blood vessels, which provide essential oxygen, nutrients, and growth factors to the bone, all of which are key factors in maintaining bone health and function [30]. Vascular injury is the initial factor in the onset of ANFH [15,16], and impaired angiogenesis is an important pathway in the occurrence of ANFH [17]. On the contrary, promoting angiogenesis and restoring the formation of blood vessels in the necrotic area to rebuild the blood supply in the necrotic area is beneficial to the collateral circulation and repair of the necrotic area.

The femoral head is supplied by vessels derived from the medial femoral circumflex artery (MFCA), inferior gluteal artery (IGA), lateral femoral circumflex artery (LFCA), obturator artery, superior gluteal artery, and first perforating branch of the deep femoral artery. Among them, the deep branch from MFCA is the most important and the main source of blood supply to the femoral head. IGA can form anastomotic with MFCA through the piriformis branch and indirectly supply blood to the femoral head. In some anatomical variations, IGA may become the main blood supply vessel of the femoral head, and more than 50% of fetuses of 16–29 weeks are thought to have IGA as the main blood supply vessel [31]. Moreover, LFCA supplies blood to the femoral neck rather than the femoral head through the anterior nutrient artery, while the superior gluteal artery, obturator artery, and the first performing branch of the deep femoral artery account for only a small part of the femoral blood supply [32].

The medial circumflex femoral artery, which is the main source of blood supply to the femoral head, is thought to be the vascular anatomy most closely associated with ANFH. However, due to the high dependence of the femoral head on blood supply, when other vessels are blocked and damaged, it also leads to insufficient blood supply to the femoral head to a certain extent, and if the blood supply cannot be restored in time, it leads to progressive death of bone cells, followed by joint surface collapse and degenerative osteoarthritis, and eventually leads to the occurrence of femoral head necrosis [33]. In addition to the above classic vessels, with the development of imaging techniques, the vascular anatomy of the femur has gradually been further studied, such as trans-cortical vessels (TCVs), and a vascular system connecting circulation within the bone marrow (BM) and periosteal circulation has been discovered [34]. And there is also a vascular subtype that mediates subchondral remodeling called H-type vessels, which have been mentioned in recent studies [35,36]. All of these studies help researchers to explain and try to treat femoral head necrosis from a new perspective.

In summary, the occurrence of ANFH is intimately associated with vascular injury. Researchers are now concerned with ways to promote vascular regeneration and repair following vascular injury brought on by a variety of causes in order to restore the femoral head’s local blood supply. Next, we will discuss the relevant mechanism and treatment measures in detail.

## 4. How to Regulate Angiogenesis to Affect ANFH

As mentioned above, angiogenesis is a process in which endothelial cells are activated from the long-term dormant quiescent state after receiving local ischemia, hypoxia, or other stimuli, which starts the angiogenesis process. Under strict control between pro-angiogenesis and anti-angiogenesis factors, the balance between pro-angiogenesis and anti-angiogenesis factors is used to generate new blood vessels, resulting in the expansion of the vascular network [37,38]. Common angiogenic factors include hypoxia-inducible fact-1α (HIF-1α) and vascular endothelial growth factor (VEGF), vascular endothelial growth factor receptors (VEGFRs), vascular endothelial cadherin (VE-cadherin), cluster of differentiation31 (CD31), delta-like typical Notch ligand 4-Notch-Noggin (DLL4-NOTCH-NOG), etc. [39]. The interaction between the molecular pathways affected by these angiogenic factors promotes angiogenesis and bone repair in necrotic areas through an angiogenesis–osteogenesis coupling and ultimately promotes the healing of femoral head necrosis [40].

The effects of these angiogenic factors on angiogenesis have been well discussed in previous reviews. Here, we will not reiterate these relevant knowledge points. We will only focus on VEGF and HIF-1α, two key angiogenic regulatory factors that have been widely studied in the pathogenesis or treatment of ANFH; other factors are briefly discussed. In addition, we also discuss the possible role of angiogenesis regulated by endothelial cell metabolism, a mechanism that has received more attention recently.

### 4.1. VEGF

At present, the most widely studied pro-angiogenic protein is vascular endothelial growth factor (VEGF). VEGF has a strong role in promoting endothelial cell growth and angiogenesis, and a large part of other angiogenic factors also play a role by enhancing the expression and generation of VEGF. VEGF signal can induce the formation of new blood vessels through vascular endothelial growth factor receptors (VEGFRs) [41]. There have been detailed descriptions of VEGF family members and their complex signaling pathways and functions [41,42]. We will not repeat this here and will mainly summarize its research in the disease model of ANFH.

Previous studies have shown that VEGF is highly expressed in necrotic areas and occurs mainly in the early stages of ANFH [43,44,45]; its main role is to promote angiogenesis and osteogenesis [46] and it also can promote endothelial cell proliferation [47]. VEGF plays an important role in the repair of hypoxia-induced osteonecrosis [48]. VEGF is upregulated under the induction of hypoxia-inducible factor 1α (HIF-1α) in the ischemic hypoxia environment. Moreover, upregulated VEGF can promote the growth of endothelial cells, increase the number and volume of blood vessels in the necrotic area of the femoral head, and restore local blood supply, thereby promoting the repair of necrotic areas of the femoral head [49,50]. In the absence of VEGF, local angiogenesis and repair of necrotic areas in femoral head necrosis are weakened. Overexpression of VEGF can promote osteogenesis and angiogenesis of AMSCs. In addition to promoting the growth and differentiation of endothelial cells and thus increasing angiogenesis, VEGF has also been shown to directly promote the recruitment of bone marrow-derived endothelial progenitor cells, thereby promoting the formation of local blood vessels [51,52]. In the absence of VEGF, local angiogenesis and repair of necrotic areas in femoral head necrosis are weakened [53], while the overexpression of VEGF can promote osteogenesis and angiogenesis [54].

VEGF has attracted the attention of many researchers in the field of stem cell treatment of ANFH (Table 1) [55,56]. Since MSCs have been shown to promote angiogenesis in vivo by inducing the release of VEGF [57], arterial perfusion of MSCs has been proposed by researchers to improve the blood supply of the femoral head to treat ANFH, which has been verified in dog models of ANFH [58]. Another three-year follow-up study on the efficacy of mesenchymal stem cells in the treatment of ANFH further demonstrated the role of MSC transplantation in the treatment of ANFH [59]. There are also some researchers who combined MSC-targeted arterial perfusion with porous tantalum [60], or directly implanted vascular endothelial growth factor 165 (VEGF165) transgenic MSCs into animal models of femoral head necrosis established by femoral neck osteotomy [61]. Alternatively, platelet-rich plasma clot releasate (PRCR) and MSCs are used together [62]. These measures can provide a positive impact on the treatment of ANFH by promoting local angiogenesis of the femoral head.

In terms of biomaterials, combining MSCs with strontium-doped calcium phosphate (SCPP) synthetic scaffolds can also improve local angiogenesis of femoral head necrosis by increasing the expression of VEGF [63]. The MSCs treated with VEGF and bone morphogenetic protein-6 (BMP-6) genes were combined with biomimetic synthetic scaffold poly lactide-co-glycolide (PLAGA) and then subcutaneously implanted into nude mice, which enhanced osteogenesis and angiogenesis in mice [64]. In terms of biological scaffolds, other researchers have prepared calcium phosphate (CPC) composite scaffolds containing poly lactic co glycolic acid (PLGA) microspheres loaded with bone morphogenetic protein (BMP) and VEGF (BMP-VEGF-PLGA-CPC) [65], or 0.25% nano-hydroxyapatite-copper–lithium (0.25% Cu-Li-nHA) scaffolds, both of which can provide the enhancement of the local VEGF pathway to promote angiogenesis to repair ANFH [66]. In addition, recently, researchers have developed an injectable hydrogel containing angiogenesis stimulator peptide (QK), which can enhance the expression of local VEGF after injection, thereby effectively enhancing angiogenesis and inhibiting ANFH [67]. Moreover, extracorporeal shock wave therapy (ESWT) and microbubble-mediated ultrasound (MUS) have also been shown to promote the upregulation of VEGF, thereby inducing neovascularization and improving blood supply to the femoral head [68,69]. Some are treated with drugs, such as desferoxamine (DFO), Thymoquinone (TQ), Tongluo Shenggu Capsule (TLSGC), and astragaloside IV (AS-IV), which have also been shown to promote osteogenesis and angiogenesis through the VEGF pathway, thereby preventing or treating ANFH [70,71,72,73]. Recently, some studies have used exosomes to increase the content of local VEGF, which also provides a new perspective for the treatment of ANFH [74,75,76].

### 4.2. HIF

HIF is a family of transcription factors believed to be the primary regulator of cellular responses to hypoxia, consisting of a heterodimer of constitutively expressed subunit HIF-β and oxygen-regulated subunit of HIF-α [77]. Under hypoxia, the hydroxylation of HIF-α is inhibited by lower oxygen levels, so HIF-α subunits are transferred into the nucleus and dimerize with the HIF-β subunits, which in turn regulate the transcription of at least more than 100 target genes (such as VEFG and erythropoietin), thereby playing a role in the ischemia and hypoxia environment [78,79,80] and affecting physiological or pathological angiogenesis [81]. Previous studies have shown that glucocorticoids can reduce the expression of HIF-1α and inhibit angiogenesis, which leads to the collapse of the femoral head and the occurrence of ANFH. Downregulation of the HIF pathway may be one of the important causes of the occurrence of ANFH [82,83,84,85]. Moreover, HIF-1α is involved in the local repair response of ANFH [45], which demonstrates the key role of HIF-1α in the disease of ANFH. Although HIF can affect the expression of many target genes, VEGF may be the main target gene of HIF-1α in ANFH [86]. As mentioned earlier, HIF-1α is a precursor of upregulation of VEGF. After HIF-1 α and transgenic bone marrow cells are transplanted into the necrotic area, VEGF is upregulated and angiogenesis increases, thus promoting the repair of femoral head necrosis [50].

In terms of specific applications, some drugs that affect HIF have been studied in models of ANFH (Table 2). For example, astragaloside IV (AS-IV) requires HIF-1α intermediation in the process of increasing local angiogenesis through VEFG [70]; moreover, desferoxamine (DFO) alone or in combination with alendronat has been shown to enhance angiogenesis by promoting HIF-1α activation, thereby playing a protective role in ANFH [73,87,88]. 3, 4-Dihydroxybenzoate (EDHB) can prevent the occurrence of ANFH by inhibiting HIF-1α degradation and increasing VEGF expression [89]. In addition, some researchers transfected bone marrow mesenchymal stem cells with adenovirus carrying triple-point mutations (amino acids 402, 564, and 803) in the HIF-1α coding sequence (CDS), and injected its derived exosomes into the necrotic area, which also promoted the repair of avascular necrosis of the femoral head through local angiogenesis [19]. Other researchers have found that hypoxia pre-stimulated bone marrow mesenchymal stem cells can express more HIF-1 α, which can better stimulate local angiogenesis and bone regeneration after transplantation [90]. Before hypoxia induction, using HIF-1 α gene transfection or lentivirus encoding HIF-1 α to infect bone marrow stem cells can further improve the curative effect [50,91], and transplanting EPC transfected with Ad-BMP-2-IRES-HIF-1α into the avascular necrosis site of the femoral head can also have similar effects [92]. In addition, in the field of biological scaffolds, 0.25% nano-hydroxyapatite–copper–lithium (0.25% Cu-Li-nHA) scaffolds have also been shown to activate HIF-1α to increase angiogenesis in necrotic areas of the femoral head through VEGF [66].

### 4.3. Other Factors

In addition to the above-mentioned factors that have been extensively studied and applied in ANFH, other angiogenic factors have also been studied in this disease model (Table 3). For example, single nucleotide polymorphisms (SNPs) of the NRP1 gene were found to be a protective factor in the occurrence of ANFH in a genetic association study, which can reduce the occurrence of ANFH [93]. In addition, the expression of platelet-derived growth factor-bb (PDGF-BB) is suppressed by hormones in steroid-induced necrosis of the femoral head, resulting in a decrease in H-type angiogenesis, which affects the local angiogenesis–osteogenesis coupling and leads to the occurrence of ANFH [84]. The use of PDGF-BB has been shown to improve the local blood flow of ANFH [94]; therefore, some researchers transfected MSCs with lentivirus of the PDGF-BB gene under the control of phosphoglycerate (PGK) to obtain PGK-PDGF-BB-MSCs and performed experiments in rabbit models of ANFH. It was found that the injection of the bone tunnel during core decompression can promote angiogenesis in the early treatment of femoral head necrosis and reduce the occurrence of ANFH [95]. And cartilage oligomeric matrix protein angiopoietin-1 (COMP-Ang1) as an angiogenesis factor directly injected into the necrosis area can promote angiogenesis, making the local angiogenesis show higher levels of vascularity [96]. The combination of bone morphogenetic protein-2 (BMP-2) and COMP-Ang1 has been found to better improve angiogenesis in the necrotic area of the femoral head, thereby protecting the femoral head [97]. In addition, metformin has been shown to maintain vascular density by inducing the expression of Ang1 [98].

As mentioned before, BMP is often used in combination with other angiogenic factors to treat ANFH, such as VEGF [64,65], [99,100,101], HIF-1α [92], COMP-Ang1 [97], and basic fibroblast growth factor (bFGF) [102], which have all been reported in combination with BMP. Researchers have also shown that DFO [87], low-intensity pulsed ultrasound (LIPUS) [103], and microbubble-mediated ultrasound (MUS) [69] can increase the local expression of BMP-2, thus improving the local angiogenesis and bone repair. Moreover, hepatocyte growth factor (HGF) can enhance the expression of BMP-2 and enhance angiogenesis in the local fracture, which may be further studied in a model of ANFH [104]. Finally, in addition to the means mentioned above, the use of platelet-rich plasma (PRP) [105], Vitamin K2 [106], shockwave treatment [107], arterial infusion of autologous liposuction cells (LPCs) [108], and block of IL-6 [109] are also considered to promote angiogenesis and play a protective role in the occurrence of ANFH.

**Table 3 biomedicines-12-02577-t003:** Related research on using other angiogenesis regulatory factors to treat ANFH.

Therapy	Methodology	Functional Output	Period Studied	Reference
Drug	Platelet-derived growth factor-bb (PDGF-BB)	Mediates the self-renewal of MSCs and maintains their osteogenic ability, stabilizing the newly formed vascular tubes by recruiting MSCs for improving intraosseous vascular integration	2023	[94]
Transgenic technology combined with stem cell therapy	Transplantation of PDGF-BB transgenic BMSCs	Enhances bone regeneration and angiogenesis in the treatment of early-stage ANFH	2021	[95]
Drug	Cartilage oligomeric matrix protein angiopoietin-1 (COMP-Ang1) alone or in combination with bone morphogenetic protein-2 (BMP-2)	Promotes angiogenesis and bone remodeling	2009, 2014	[96,97]
Drug	Metformin	Induces the expression of Ang1	2020	[98]
-	Bone morphogenetic protein in combination with other pro-angiogenic factors (VEGF, HIF-1α, COMP-Ang, basic fibroblast growth factor (bFGF))	Promotes osteogenesis and angiogenesis	2010–2018	[64,65,92,97,99,100,101]
Drug	Deferoxamine (DFO)	Increases the expression of HIF-1α, VEGF, BMP-2, and OCN to improve angiogenesis and bone repair	2015	[87]
Drug	Vitamin K2	Increases the level of angiogenesis-related proteins and enhances angiogenesis	2016	[106]
Physiotherapy	Low-intensity pulsed ultrasound (LIPUS)	Promotes the increase in BMP-2 and VEGF expression, thereby enhancing osteogenesis, neovascularization, and the biomechanical strength of the femoral head	2015	[103]
Physiotherapy	Microbubble-mediated ultrasound (MUS)	Promotes the increase in BMP-2, thereby enhancing osteogenesis and angiogenesis	2018	[69]
Physiotherapy	Shockwave treatment	Increases the levels of VEGF, FGF, and vWF and promotes angiogenesis	2011	[107]

### 4.4. EC Metabolism

EC metabolism is another key regulatory factor of angiogenesis, which has received more and more researchers’ attention in recent years. Considering that most metabolic enzymes are druggable, EC metabolism has special value for the development of new therapies [110]. In EC metabolism, because the mitochondria of endothelial cells are only 2–5% of the cytoplasmic volume, oxidative phosphorylation is not considered to be the main metabolic pathway of endothelial cells, and glycolysis is the main metabolic pathway of ATP production, accounting for about 85% of ATP production [111,112]. Other ways of metabolism include the peculiar use by angiogenic ECs of FA oxidation for nucleotide synthesis and of glutamine for tricarboxylic acid (TCA) cycle anaplerosis and asparagine synthesis, which have been explained in detail in related reviews [112].

Disruption of EC metabolism can lead to impaired angiogenesis, pathological angiogenesis, or vascular defects [113,114,115,116]. At present, although there is no direct evidence to show whether the disruption of EC metabolism plays a key role in the occurrence of ANFH, transcriptome studies on the local bone tissue of patients with ANFH showed that the differentially expressed genes (DEGs) between the control group and the necrosis group mainly influenced the PI3K-Akt pathway, and were mainly involved in the glycolysis/gluconeogenesis pathway [117]. In addition, studies have shown that the glycolysis and TCA cycles in patients with ANFH are significantly affected [118]. All these indicate that ANFH will destroy the local normal EC metabolism. We can guess that the destruction of normal EC metabolism will also affect the local angiogenesis of the femoral head and lead to the occurrence of ANFH, which may be a new direction in the field of ANFH.

## 5. Discussion

Angiogenesis refers to the process of existing blood vessels developing into new blood vessels. Angiogenesis is a complex process, which is regulated by a variety of pro-angiogenesis and anti-angiogenesis factors. Under disease conditions, the vascular system either undergoes excessively pathological growth, becomes dysfunctional, or is damaged, rendering it unable to respond to ischemia and hypoxia conditions and meet the local blood supply demand, leading to the occurrence of disease. In ANFH, as the longest bone in the human body, the vascularization of the femur is extremely high, and its mechanical strength, performance, repair, regeneration, and remodeling ability all depend on the health of blood vessels [30]. If effective treatment is not carried out in the early stage, this cell death will inevitably lead to the collapse of the femoral head and subsequent osteoarthritis. At present, the exact pathophysiological mechanism behind ANFH is not always clear and is generally considered to be multifactorial [119], but no matter what the inducing factor is, the result is basically the death of bone cells and bone marrow due to insufficient blood flow of the proximal femoral subchondral bone [120]. Vascular injury is the initial factor of ANFH [15,16], and impaired angiogenesis is an important pathway in the occurrence of avascular necrosis of the femoral head [17]. If not treated effectively at an early stage, this cell death will inevitably lead to the collapse of the femoral head and subsequent osteoarthritis, and eventually lead to the occurrence of femoral head necrosis.

Therefore, early diagnosis and timely regulation of angiogenesis of the femoral head play an important role in the treatment and prevention progression of ANFH. However, patients with osteonecrosis of the femoral head are often asymptomatic in the early stage, and its pathological features are often similar to cysts or lesions in subchondral bone, vasculitis, and transient osteoporosis of the hip or osteoarthritis. The inconspicuous symptoms and uncharacteristic early pathological manifestations of ANFH make its early diagnosis difficult. Therefore, clinical workers should pay attention to the imaging diagnosis of ANFH. When patients with osteonecrosis of the femoral head have symptoms, their medical history is usually hip pain, which is aggravated when walking or climbing stairs, and there are symptoms such as limited local activity, abduction, internal rotation pain, and pain in the palpation hip area [119,121]. When encountering these symptoms, doctors should pay special attention to the patient’s history and make a diagnosis in combination with an imaging examination (the gold standard is MRI); the “crescent sign” is a typical sign of early femoral head necrosis. Clinicians should pay special attention to imaging signs and stages in the early stage and give corresponding treatment in time [2,119].

After diagnosis, patients with ANFH should be treated early in time; otherwise, surgical joint replacement is the only feasible treatment measure in the later stage. In this review, we focused on the treatment of ANFH by pro-angiogenesis and discussed some key molecules in the formation of blood vessels, especially HIF-α and VEGF. We also briefly combed other pro-angiogenesis factors, and we sorted out the existing measures for the treatment or prevention of femoral head necrosis using these targets and explored the potential of some of them as therapeutic targets. Among these factors, VEGF is the most important pro-angiogenic factor, which can promote endothelial cell proliferation and angiogenesis in the local area of the ANFH. While HIF-1α is often regarded as an upstream factor regulating VEGF expression, many treatments are used to activate pro-angiogenesis through the HIF-1α/VEGF signaling pathway to treat ANFH. As for other pro-angiogenic factors, except that BMP is often mentioned in combination with other pro-angiogenic factors (including VEGF and HIF-1α) and has been widely studied, the application of other pro-angiogenic factors has mostly been mentioned in a few articles. On the basis of the study of the molecular mechanism of these pro-angiogenic factors in vivo, drugs, stem cell therapy, transgenic technology, physiotherapy, biological scaffolds, and other means are used to affect their activation in vivo, so as to promote local angiogenesis of a necrotic femoral head. Among these therapeutic measures, drug therapy is the most classic means, while stem cell therapy has gained more and more attention in recent years, and it is often used in combination with arterial perfusion therapy. Each of these treatments has advantages and disadvantages. Future observational and interventional trials will further analyze these treatments and clarify which treatment is most appropriate at which stage. In addition, we also analyzed the recently popular EC metabolism factor and discussed its possible role in femoral head necrosis, which may be a new direction for research and treatment.

Except for the parts we highlight, due to the complexity of ANFH etiology and the influence of genetic traits on angiogenesis, the interaction of gene variants with local environmental factors may increase the severity of ANFH, and response to specific treatments is closely related to an ANFH patient’s genetic endowment [122]. Some researchers suggest that genetic hotspots and their responses to different recommended ANFH treatments should be studied to reveal those predictive markers based on polygenic risk scores. Moreover, treatment plans should be designed according to the unique genotypes of ANFH patients and their responses to relevant treatment plans, so as to achieve accurate and individualized treatment of ANFH. In addition, other researchers used single-cell RNA sequencing to explore the single-cell transcriptome characteristics of ANFH [7], or integrate transcriptomic and proteomic methods to screen for differentially expressed genes (DEGs) and differentially expressed proteins (DEPs) between the control and ANFH groups [117]. All of these methods provide a new perspective for the study of the pathogenesis of ANFH, which is conducive to the design of new treatments for ANFH.

## 6. Conclusions

In this review, to better understand the mechanism and treatment of ANFH, we focused on the treatment of ANFH by pro-angiogenesis and discussed some key molecules in the formation of blood vessels. We finally conclude that providing alternative pathways through angiogenesis to resolve the problem of the obstructed free flow of the blood is an important means of treating ANFH. VEGF and HIF-α are the two most important pro-angiogenic factors in this process; other angiogenic factors have also been studied to varying degrees in models of ANFH. Except for pro-angiogenic factors, miRNA, exosomes, and EC metabolism have also been validated as novel pathways for regulating angiogenesis in many disease models, and may provide a new perspective in the treatment of ANFH. Recently, some researchers have studied local DEGs or DEPs in patients with ANFH through transcriptomic and proteomics methods, so as to further explore the pathogenesis and treatment measures of ANFH, which may help clinicians to conduct more accurate treatment of ANFH in the future. Researchers need to further analyze these treatment methods through observational and intervention trials in the future, and clarify which treatment method is most suitable at which stage.

## Figures and Tables

**Figure 1 biomedicines-12-02577-f001:**
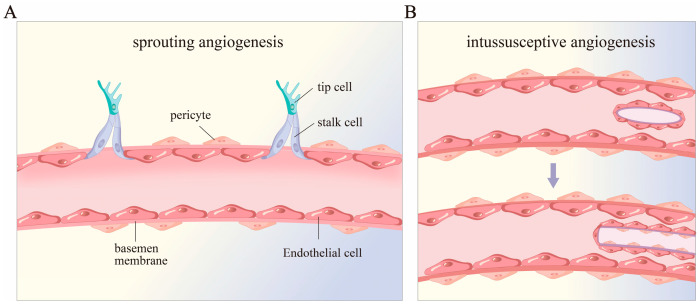
Different forms of angiogenesis. (**A**) sprouting angiogenesis; (**B**) intussusceptive angiogenesis (IA, aka vessel splitting or non-SA).

**Table 1 biomedicines-12-02577-t001:** Related research on using VEGF to treat ANFH.

Therapy	Methodology	Functional Output	Period Studied	Reference
Stem cell therapy	MSC arterial perfusion	Promotes VEGF expression	2012	[58]
Stem cell therapy combined with mechanical support therapy	Arterial perfusion of G-CSF-stimulated peripheral blood stem cells (PBSCs) combined with mechanical support therapy	Enhances the efficacy of biomechanical support in the treatment of ANFH	2015	[60]
Transgenic technology combined with stem cell therapy	Transplant VEGF transgenic bone marrow mesenchymal stem cells	Enhances bone reconstruction and vascular regeneration of the ANFH model	2012	[61]
Stem cell therapy combined with platelet-rich plasma clot releasate (PRCR)	Ultrasound-guided injection of PRCR+ umbilical cord mesenchymal stem cells (UC-MSCs)	Promotes osteogenesis and angiogenesis; suppresses osteoclast overactivity	2022	[62]
Biomaterials combined with cell therapy	Synergism between strontium-doped calcium polyphosphate (SCPP) and autologous bone marrow mononuclear cells (BM-MNCs)	Promotes osteogenesis and angiogenesis, allowing consolidation and remodeling into new trabecular bone	2015	[63]
Transgenic technology combined with stem cell therapy	Transplant VEGF and BMP6 transgenic bone marrow mesenchymal stem cells	Promotes osteogenesis and angiogenesis	2018	[64]
Biological scaffold	Calcium phosphate (CPC) composite scaffold containing poly lactic co glycolic acid (PLGA) microspheres loaded with bone morphogenetic protein (BMP) and VEGF (BMP-VEGF-PLGA-CPC)	Improves the therapeutic effect of core decompression surgery; promotes osteogenesis and angiogenesis	2016	[65]
Biological scaffold	0.25% nano-hydroxyapatite–copper–lithium (0.25% Cu-Li-nHA) scaffolds	Upregulates the HIF-1α/VEGF pathway, which benefits the repair of ANFH	2021	[66]
Biological scaffold	Local injection hydrogel containing angiogenesis stimulator peptide (QK)	Promotes the proliferation and differentiation of BMSCs and endothelial cells; promotes osteogenesis and angiogenesis	2024	[67]
Physiotherapy	Extracorporeal shock wave therapy (ESWT) and microbubble-mediated ultrasound (MUS)	Promotes the upregulation of VEGF, thereby inducing neovascularization	2007, 2018	[68,69]
Drug	Desferoxamine (DFO), Thymoquinone (TQ), Tongluo Shenggu Capsule (TLSGC), astragaloside IV (AS-IV	Promotes osteogenesis and angiogenesis through the VEGF pathway	2020–2023	[70,71,72,73]

**Table 2 biomedicines-12-02577-t002:** Related research on using HIF to treat ANFH.

Therapy	Methodology	Functional Output	Period Studied	Reference
Drug	Astragaloside IV(AS-IV)	Promotes osteogenesis and angiogenesis; inhibits cell apoptosis through the Akt/HIF-1 α/VEGF signaling pathway	2023	[70]
Drug	Desferoxamine (DFO) alone or in combination with alendronat	Increases HIF-1α expression in ANFH; promotes angiogenesis, osteogenesis, and bone repair	2020, 2015, 2020	[73,87,88]
Drug	3, 4-Dihydroxybenzoate (EDHB)	Inhibits HIF-1α degradation, promotes angiogenesis, and inhibits the apoptosis of bone cells and hematopoietic tissues	2014	[89]
Exosome	Exosomes secreted from mutant-HIF-1α-modified bone marrow-derived mesenchymal stem cells	Promotes bone regeneration and angiogenesis; increases bone trabecular reconstruction and microvascular density	2017	[19]
Stem cell therapy	Hypoxia pre-stimulated bone marrow mesenchymal stem cells (BMSCs)	Hypoxia pre-stimulation can make BMSCs express more HIF-1 α, which can better stimulate local angiogenesis	2021	[90]
Transgenic technology combined with stem cell therapy	Transplantation of HIF-1α transgenic BMSCs	Promotes osteogenesis and angiogenesis	2013, 2022	[50,91]
Transgenic technology combined with stem cell therapy	Transplantation of Ad-BMP-2-IRES-HIF-1αmu transgenic EPCs	Promotes osteogenesis and angiogenesis	2017	[92]
Biological scaffold	0.25% nano-hydroxyapatite–copper–lithium (0.25% Cu-Li-nHA) scaffolds	Upregulates the HIF-1α/VEGF pathway, which benefits the repair of ANFH	2021	[66]

## Data Availability

Not applicable.

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
