# Peer review of "Angiogenesis of Avascular Necrosis of the Femoral Head: A Classic Treatment Strategy"

_biomedicines, 2024, doi:10.3390/biomedicines12112577_

Round 1
Reviewer 1 Report
Comments and Suggestions for Authors
The manuscript entitled “Angiogenesis of Avascular Necrosis of Femoral Head: A Classic Treatment Strategy.”, In this review, the authors briefly introduced the process of angiogenesis and the blood supply to the femoral head, with a focus on summarizing the existing research on promoting angiogenesis for the treatment of ANFH.
Below are some suggestions:
In the Abstract:
The abstract is well-written, but I suggest the authors include the main results and a conclusion.
1. In the Introduction:
- The introduction is clear and concludes with the aim of the review.
2. Angiogenesis and 3. Vascular distribution of femoral head
- These two items are well described, but the authors should insert at least one image to better visualize the two processes.
4. How to regulate angiogenesis to affect ANFH
- As this is a review manuscript, I suggest that the authors insert a brief paragraph with the research data: database, period studied, number of articles......including a heregrogram could make it easier to see this data.
5. In the Discussion:
- The authors should rewrite the discussion, which is too concise: they should compare the results of the articles analyzed and not just describe the processes again as they did previously.
6. Conclusion:
- The conclusion should also be rewritten, stating the main objective of the manuscript, the main findings, a final conclusion and future prospects.
* Authors should adjust their references according to the journal's guidelines.
Author Response
Comment1: In the Abstract:The abstract is well-written, but I suggest the authors include the main results and a conclusion. Reply1: Thank you for your suggestion. We have added the main conclusion and outlook sections to the abstract, hoping to help readers better understand the content of our article through the abstract section (Line:11-22). Comment2: Angiogenesis and 3. Vascular distribution of femoral headThese two items are well described, but the authors should insert at least one image to better visualize the two processes. Reply2: Thank you for your suggestion. In previous versions, missing images did indeed make the article too monotonous and unable to express the content intuitively. We have added an image to the Angiogenesis section as per your suggestion, to describe sprouting angiogenesis and intussusceptive angiogenesis (IA, aka vessel splitting or non-SA), two classic methods of angiogenesis, in order to provide readers with a more intuitive understanding of these two sections (see in figure1). Comment3: How to regulate angiogenesis to affect ANFHAs this is a review manuscript, I suggest that the authors insert a brief paragraph with the research data: database, period studied, number of articles......including a heregrogram could make it easier to see this data. Reply3: Thank you very much for your suggestions. We have carefully considered your suggestions and marked the publication period of each relevant research paper on the original Table 1-3, so as to facilitate readers to understand the research content focused on in each period (See in Table 1-3). Comment4: In the Discussion:The authors should rewrite the discussion, which is too concise: they should compare the results of the articles analyzed and not just describe the processes again as they did previously.

Reviewer 2 Report
Comments and Suggestions for Authors
The review by P.Wang et al presents information about the current state and prospects of prevention and treatment of Avascular Necrosis of Femoral Head (ANFH) using approaches to enhance tissue angiogenesis. The authors describe the anatomy of the vascular network of femoral head, indicate the most important highways supplying the tissue with nutrients. Among the main molecular factors of femoral head angiogenesis HIF1-alpha, VEGF/VEGFR pathway, VE-Cadherin and Notch signaling are mentioned. Attention is also paid to the use of cell therapy in combination with angiogenesis factors and scaffold-factors. Separate chapters of the review are devoted to the use of VEGF, HIF-1 alpha, some other factors used by a number of researchers in the treatment of ANFH. The peculiarities of endothelial cell metabolism characterized by predominance of glycolysis over mitochondrial oxidative phosphorylation are also considered.
Based on the set of information presented in the work, the authors are inclined to the opinion that the combined application of various angiogenic factors with cell therapy (in particular, using exosomes) with regulators of energy metabolism is promising.
The reviewer has no major comments to this work.
Minor points.
It is highly desirable to summarize the information given in the review in the form of a graphical diagram or figure.
Line 63: ‘The vascular system is a closed tree-shaped structure network composed mainly of endothelial cells (ECs), which are formed by arteries, veins and capillaries...’: it is desirable to indicate that vascular tissue also includes smooth muscle cells, fibroblasts, pericytes and mesenchymal stromal cells close to them.
Lines 142-144: Text needs some correction: ‘vascular endothelial growth factor receptors (VEGFR), Vascular endothelial growth factor receptors (VEGF), Vascular endothelial cadherin (VEGFR) VE-cadherin, cluster of differentiation31, CD31)...’.
Line 168: hypoxia-induciblefaction-1α (HIF-1α)
Table 1: inhibitting osteoclast overactivity
Line 277: As we have mentioned many times before

Author Response
Comment1:Line 63: ‘The vascular system is a closed tree-shaped structure network composed mainly of endothelial cells (ECs), which are formed by arteries, veins and capillaries...’: it is desirable to indicate that vascular tissue also includes smooth muscle cells, fibroblasts, pericytes and mesenchymal stromal cells close to them.
Reply1:Thank you very much for your suggestion. In this part, we wanted to emphasize that blood vessels are mainly composed of endothelial cells, so we omitted the description of other cells that make up blood vessels. After your prompt, we found that this may not be appropriate, so we have supplemented this section in accordance with your suggestions. (Line63-65)
Comment2:Lines 142-144: Text needs some correction: ‘vascular endothelial growth factor receptors (VEGFR), Vascular endothelial growth factor receptors (VEGF), Vascular endothelial cadherin (VEGFR) VE-cadherin, cluster of differentiation31, CD31)...’.
Reply2:Thank you for your careful review, this part is our negligence in writing, leading to some text duplication and abbreviation errors, we have corrected accordingly.(Line 144-147)
Comment3:Line 168: Text needs some correction: hypoxia-induciblefaction-1α (HIF-1α)
Reply3: I am very sorry for making such a simple mistake as the misspelling of HIF-1α. Corresponding changes have been made in the article(Line170)
Comment4:Table 1: Text needs some correction: inhibitting osteoclast overactivity
Reply4:Thank you for your careful review, this part is really not clear enough, we have revised the corresponding content in Table 1.
Comment5:Line 277: Text needs some correction:As we have mentioned many times before
Reply5:I'm really sorry that there are so many low-level errors in the grammar, we have corrected them and re-checked the grammar of the whole article.(Line279)

Reviewer 3 Report
Comments and Suggestions for Authors
Dear Author,
the review illustrates femoral head necrosis, providing information about the mechanism and potential strategies for more effective use of angiogenesis
However In my opinion are not present paragraphs on interesting findings such as a transcriptomic profile of cells involved in osteonecrosis and the potential role of miRNAs as markers of disease progression or response to therapy
Comments on the Quality of English LanguageThe English need a revision
Author Response
Comment1: However In my opinion are not present paragraphs on interesting findings such as a transcriptomic profile of cells involved in osteonecrosis and the potential role of miRNAs as markers of disease progression or response to therapy
Reply1:Thank you very much for your suggestion. The two perspectives you mentioned were indeed not taken into consideration when we wrote the article. When designing this review, we chose to focus on the role of angiogenic factors in ANFH and briefly introduce the function and therapeutic prospects of endothelial metabolism. While the transcriptomic profile of cells involved in osteonecrosis and the potential role of miRNAs as markers of disease progression or response to therapy were not discussed. This was because we considered that introducing too much content might make the article less focused, and the topic of our article was the association between angiogenesis and ANFH.However, after receiving your advice, we carefully considered the structure of the article and searched for relevant content, and believe that it is indeed necessary to provide some introduction and discussion. Therefore, in the discussion section, we analyzed the viewpoints you mentioned, briefly introduced the articles related to angiogenesis, and provided some prospects(Line:375-388).Thank you again for your valuable advice.

Round 2
Reviewer 1 Report
Comments and Suggestions for Authors
No comments.